# RSONAR: Data-Driven Evaluation of Dual-Use Star Tracker for Stratospheric Space Situational Awareness (SSA)

**DOI:** 10.3390/s26010179

**Published:** 2025-12-26

**Authors:** Vithurshan Suthakar, Ian Porto, Marissa Myhre, Aiden Alexander Sanvido, Ryan Clark, Regina S. K. Lee

**Affiliations:** 1Department of Earth and Space Science, York University, Toronto, ON M3J 1P3, Canada; ianporto@my.yorku.ca (I.P.); reginal@yorku.ca (R.S.K.L.); 2Department of Electrical and Computer Engineering, McMaster University, Hamilton, ON L8S 4K1, Canada; 3MDA Space, 13800 Commerce Pkwy, Richmond, BC V6V 2J3, Canada

**Keywords:** space situational awareness (SSA), space domain awareness (SDA), resident space objects (RSOs), stratosphere, star tracker, optical imaging

## Abstract

The growing density of Earth-orbiting objects demands improved Space Situational Awareness (SSA) to mitigate collision risks and sustain space operations. This study demonstrates a dual-purpose star tracker (ST) for SSA using data from the Resident Space Object Near-space Astrometric Reconnaissance (RSONAR) stratospheric balloon campaign under the 2022 Canadian Space Agency–Centre National d’Études Spatiales (CSA–CNES) STRATOS program. The low-cost optical payload—a wide-field monochromatic imager flown at 36 km altitude—acquired imagery subsequently used for post-processed attitude determination and Resident Space Object (RSO) detection. During stabilized pointing, over 27,000 images yielded sub-pixel astrometry and stable image quality (mean full-width-Half-maximum ≈ 388 arcsec). Photometric calibration to the Tycho-2 catalog achieved 0.37 mag root mean square (RMS) scatter, confirming radiometric uniformity. Apparent angular velocities of $7\times 10^{2}$ to $8\times 10^{3}$ arcsec $s^{-1}$ corresponded to sunlit low-Earth-orbit (LEO) objects observed at 25°–35° phase angles. Covariance-weighted Mahalanobis correlation with two-line elements (TLEs) achieved sub-arcminute positional agreement. The Proximity Filtering and Tracking (PFT) algorithm identified 22,036 total RSO and 387 total streaks via image stacking. Results confirm that commercial off-the-shelf STs can serve as dual-use SSA payloads, and that stratospheric ballooning offers a viable alternative for optical SSA research.

## 1. Introduction

Since the launch of Sputnik on 4 October 1957, satellites have become an integral part of modern life. As of May 2024, approximately 119 space agencies and countries operate nearly 9000 active satellites in orbit [1]. With companies like Starlink deploying mega-constellations, these figures are expected to rise. Recent orbital-launch statistics further underscore this growth: global launch activity has accelerated sharply in the past decade, reaching a record 263 launches in 2024—led by the United States (145) and China (68) [2]. In addition to active satellites, numerous debris fragments, defunct payloads, and rocket bodies populate near-Earth space. The Low Earth Orbit (LEO) region remains the most congested, containing 57% of monitored RSOs, with 9016 space debris originating from rockets and payloads [3]. These include fragments from rocket bodies, dislodged components from decaying satellites, and debris from collisions and anti-satellite weapons (ASAT) tests. These objects travel at hypervelocities, typically 7–8 km/s in LEO [4], posing a significant threat to operational satellites [5]. The accidental collision between Iridium 33 and Cosmos-2251 on 10 February 2009, generated 2294 debris [5]. Similarly, the ASAT test of Fengyun-1C on 11 January 2007, resulted in 3431 RSOs [5]. Consequently, it is increasingly necessary to identify, track, and catalog RSOs as part of space situational awareness (SSA) and space domain awareness (SDA) efforts to prevent collisions and safeguard space missions from premature termination [6,7,8].

### 1.1. Overview of Ground and Space-Based SSA Architecture

Traditionally, SSA has been conducted using ground-based lasers, radars and optical sensors [9,10,11,12,13,14,15]. The sensors may be classified as either ‘active’ or ‘passive’, with the observational periods and operations constrained by the instrumental setup. However, SSA from ground-based sensors is limited by their geographic locations, as there are inherent variations in sensor distribution between the northern and southern hemispheres. Consequently, previously tracked objects may move out of the sensors’ reach for months when their orbits shift, thereby affecting traceability [16]. Although a single sensor can acquire tracks, incorporating temporal and geographic diversity (i.e., multi-pass and multi-site coverage) substantially improves orbit-determination conditioning, strengthens measurement association, reduces coverage gaps, and thereby enhances catalog maintenance [9,10,12]. Single-pass optical tracks are poorly conditioned for initial orbit determination, whereas combining observations acquired at different times or from varied viewing geometries markedly improves association and state estimation [17]. Radar systems become less effective at detecting objects as distance increases because the strength of the returned signal diminishes rapidly. Specifically, the received signal power decreases proportionally to the fourth power of the distance to the target. To compensate, radar systems may require higher transmission power, larger antennas, or more sensitive receivers, but these enhancements have practical limitations. While many ground-based radars, particularly phased array systems, are optimized for detecting objects in LEO and Medium Earth Orbit (MEO) [16], certain radar systems—such as C-band radars and advanced commercial systems like Thoth Technology’s Earthfence—are capable of tracking objects in Geosynchronous Earth Orbit (GEO) [18]. In contrast, ground-based telescopes can detect objects in higher orbits such as GEO. However, the operational period of these telescopes is limited by cloud coverage, and they are only functional during the night. These systems are also susceptible to light pollution, sky brightness and exhibit reduced performance when imaging RSOs located within approximately 15° of the Moon’s center [19]. Furthermore, the efficacy of telescopes is determined by the illumination of the object, its size, reflectivity and the light collection properties of the telescope itself. Additionally, images captured by these telescopes are significantly influenced by atmospheric turbulence and light pollution. Sensors in low-elevation regions experience diminished observational capabilities due to the additional atmospheric presence. Currently, ground-based optical sensors serve as the primary source of optical data for RSO identification and characterization. The detection of RSOs in LEO using ground-based images presents a significant challenge due to the telescopes’ limited field of view (FOV) as the RSOs transit at hypervelocities. Nevertheless, GEO observation is feasible because these satellites remain fixed relative to the Earth’s surface and within the FOV of ground-based sensors. To maintain tracking, the sensors must operate in Track Rate Mode (TRM) on a precision motorized platform.

A key advantage of space-based sensors is their ability to observe objects without interference from Earth’s atmosphere. Space-borne imagers excel at observing objects near the sun because they operate above the Earth’s atmosphere, thus avoiding the scattering of sunlight that obscures ground-based views, especially at low solar angles. This vantage point also provides a dark celestial background, enhancing the visibility of faint objects near the bright sun. While both types of imagers have solar exclusion zones to protect their instruments, space-based sensors generally have smaller ones, allowing them to observe closer to the sun. Furthermore, they have less latency for targeted observation, which can be accomplished within a few hours. Despite these advantages, space-based systems often incur higher costs and reduced sensitivity compared to ground-based systems. According to [20], the cost (USD) of ground-based sensors ranges from $0.76 to $1.59 per observation, while that of space-based sensors ranges from $ 21.92 to $ 24.35 per observation. However, the ORS-5 satellite, observing the GEO belt, is an outlier among space-based sensors, costing $ 3.07 per observation. These values follow the cost model in [20], which annualizes system acquisition cost over the assumed lifetimes (40 years for ground-based, 7–10 years for space-based) and divides by the yearly number of observations. These figures emphasize the performance-cost trade-off between ground and orbital assets, motivating intermediate approaches such as high-altitude stratospheric platforms.

### 1.2. Dual-Use of SSA Sensor

Given the sensitivity of SSA and the constraints on data acquisition by dedicated SSA sensors, access to data is limited and rarely available to the public. A study has been conducted on the utilization of existing space-borne sensors for SSA purposes [21]. Additionally, Star Trackers (ST) are traditionally employed for attitude determination and have been investigated for RSO imaging [22,23]. ST offer a more cost-effective alternative to dedicated SSA sensors. Their wide FOV enables increased coverage, facilitating earlier detection and warning of potential conjunctions. However, Star Trackers have inherent limitations, such as lower resolution, limited sensitivity, and smaller apertures, which result in reduced imaging capabilities compared to dedicated SSA sensors. Nevertheless, their imaging is sufficient for astrometric and photometric analysis, thus providing additional information about RSOs.

### 1.3. Overview of Stratospheric Ballooning for Scientific Observations

Since the early 19th century, scientific ballooning has evolved into a significant venue for conducting cost-effective research [24]. The platform provides stable photometry, the potential for diffraction-limited imaging, and a very low sky background, enabling observations that cannot be ground-based. Additionally, due to its suborbital nature, it has been utilized for prototyping satellite designs and space experiments. According to [25], balloon-borne optical systems can match spaceborne imaging quality, as stratospheric altitudes (35 km) lie above 99% of the atmosphere, minimizing turbulence and sky brightness. Under these conditions, Fried parameter $r_{0}$ (the Fried parameter $r_{0}$ is a measure of the atmospheric coherence length and quantifies the strength of optical turbulence [26]), which quantifies atmospheric seeing, can exceed 250 m at 500 nm, enabling diffraction-limited resolution comparable to that of space-based and advanced ground-based telescopes [25].

To fulfill the scientific requirements, various types of flights and balloons have been developed. Turn-around period flights capitalize on the cessation of high-altitude winds that prevail in late spring and summer to maintain the platform’s relative stability. These flights are relatively brief, lasting a few days after launch, and payloads are retrieved in close proximity to the launch station. In contrast, transoceanic flights are long-duration flights that can last for weeks and cover vast distances. An example of such a flight is the transatlantic flight, which traverses the Atlantic Ocean from Europe to the United States [24]. Ultra-long duration balloon flights are equipped with super-pressure balloons that can remain aloft for approximately 100 days [27]. Smaller high-altitude balloons, referred to as meteorological balloons, have also been utilized for low-cost prototyping and rapid testing. While they can reach high altitudes, their payload capacity is limited to a few kilograms with short observation periods. Additionally, the flexibility of ballooning allows for payload retrieval and rapid validation of findings and performance before reflight or a space launch. Despite the advantages of ballooning, it is constrained by the risk of payload damage upon landing, which must be considered. Stratospheric flights thus serve as a critical testbed for validating system performance without requiring a full orbital launch, enabling the attainment of Technology Readiness Level (TRL) 7 through demonstration in a relevant environment. This milestone is particularly significant, as TRL 7 marks the transition from an experimental prototype to a flight-ready system, preceding flight qualification (TRL 8) and orbital validation (TRL 9).

### 1.4. Resident Space Object Near-Space Astrometric Research (RSONAR) Overview

The scientific objectives of the Resident Space Object Near-space Astrometric Research (RSONAR) mission were threefold: firstly, to demonstrate the concept of a dual-purpose ST for SSA; secondly, to acquire data with different sky backgrounds in the suborbital environment for SSA research; and thirdly, to evaluate the stratosphere as an alternative platform for SSA. Hence, a low-cost CubeSat-scale payload was developed. RSONAR was a 2U CubeSat equipped with a commercial-grade ST, a scientific Complementary Metal-Oxide-Semiconductor (sCMOS) imager, and a Field Programmable Gate Array (FPGA)-based onboard computer. The RSONAR was launched as part of the STRATOS 2022 program, a joint venture between the CSA and the Centre National d’Études Spatiales (CNES) that uses stratospheric balloons with stabilized, precise pointing period flights. Additional information on hardware design concepts, operations and payload development can be found at [28]. The dataset obtained from the flight significantly aided in the development of robust RSO detection algorithms, as it contained RSOs with different characteristics, posed challenges associated with a moving observer, and included varying illumination conditions [29]. Ongoing studies are using this dataset for AI-based RSO detection [30], and tracking.

The objective of this study is to validate the dual-purpose functionality of a commercial ST for SSA through a comprehensive analysis of the astrometric, photometric, and catalog correlation performance of stars and RSOs observed during the RSONAR stratospheric mission. In addition to evaluating attitude determination and image-based RSO detection, the study implements a covariance-weighted Mahalanobis correlation framework to associate measured topocentric positions with TLE predictions, assessing the feasibility of accurate statistical matching from a high-altitude platform. The broader goal is to establish the viability of stratospheric balloon flights as recoverable, low-cost alternatives for SSA data acquisition. Section 2 describes the RSONAR flight profile and observation geometry; Section 3 details the photometric, astrometric, and correlation procedures used to evaluate the ST’s dual capability; Section 4 and Section 5 presents the results and discussion; and Section 6 concludes with implications and future directions for stratospheric SSA observations.

## 2. RSONAR Dataset

Passive observation and imaging, without accounting for the motion of the observer and RSOs, can be referred to as Star Stare Mode (SSM). SSM employs lower exposure rates, which are often found in STs. In this mode, RSOs transiting within the imager’s FOV may appear similar to stars. Although the higher exposure rates for RSO imaging offer significant improvements in observing dimmer RSOs [31], rendering them as streaks, this technique is often utilized in TRM. Due to the nature of the stratospheric platform and RSONAR’s mission concept, TRM operation is not feasible. Consequently, imaging during the flight was conducted using SSM, which further validated the concept of dual-use STs for SSA. The RSONAR payload is shown in Figure 1, integrated onto the CSA gondola prior to flight.

On 21 August 2022, during a day–night turnaround flight, RSONAR acquired 95,046 images with an exposure rate of 0.1 s. The images were collected in bursts, with a 4 s delay between each burst, ensuring continuous data acquisition throughout the entire operation period. The payload was powered off after 537 min of operation. Upon reaching an altitude of 36 km at 01:56 a.m. local time (UTC–4), the gondola commenced a coasting phase, with ballasts being released intermittently throughout the mission to maintain stability. The optical and detector characteristics of the payload are summarized in Table 1.

The optimal period for SSA imaging occurred between astronomical dawn (4:02 a.m.) and sunrise (06:03 a.m.). During this period, the Sun’s geometric center rose from 18° below the horizon, providing sufficient illumination of RSOs without saturating the imager. This overlapped with the stabilized pointing period (SPP), which lasted from 2:56 a.m. to 5:33 a.m. The images captured shortly after the SPP were susceptible to stray light stemming from Earth’s albedo, with complete saturation occurring at 6:34 a.m. Consequently, the data obtained during the 157-min SPP represent the primary focus of analysis in this paper.

## 3. Methodology

This section outlines the methods used to assess image quality, perform astrometric and photometric calibration, and detect RSOs within the dataset. Payload attitude was determined using Astrometry.net (https://nova.astrometry.net/, accessed on 15 May 2025) [32], and image quality was quantified through full width at half maximum (FWHM) analysis of stellar point-spread functions. Photometric calibration for stellar sources employed empirical fitting to derive the zeropoint, slope, and vignetting parameters, while a separate physics-based throughput model provided independent absolute magnitude estimates for the RSOs. Astrometric precision was evaluated against catalog references, and object trajectories were correlated with TLE predictions using a Mahalanobis distance criterion. RSO detection combined streak detection with temporal stacking [29]. The major steps of the analysis pipeline, including attitude determination, photometric and astrometric calibration, and RSO detection and correlation, are summarized in Figure 2.

### 3.1. Attitude Determination

Astrometry.net was utilized to determine the attitude of the payload due to its 99.9% success rate [32]. The robust nature of Astrometry.net eliminates the need for image-pointing orientation and scale and renders optimal for passive observation on moving observer datasets, such as that of RSONAR. The application generates a list of calibration values, including Right Ascension (RA), Declination (DEC), and orientation east of north with respect to the boresight of the optical sensor. In accordance with the internal convention, yaw is defined as the RA, pitch as the negative of the DEC, and roll as the orientation east of north. For subsequent astrometric and photometric analyses, all attitude solutions were referenced in the Earth-Centered Inertial (ECI) frame and transformed into the local topocentric frame using GPS-derived balloon coordinates; image-plane positions were then projected via the World Coordinate System (WCS) transformation to ensure consistent alignment between celestial, observer, and detector reference frames.

Gondola position and state information were provided by the CSA μPRISM avionics module. μPRISM supplied GPS latitude, longitude, and altitude at a 1 Hz rate through standard NMEA sentences, with differential-GPS refinement used for attitude and heading. The geodetic coordinates were reported to four decimal places in latitude and longitude (corresponding to a numerical resolution of ≈10–12 m at mid-latitudes) and 1 m in altitude. These GPS measurements were used directly as the observer position for all subsequent astrometric and photometric transformations.

### 3.2. Full Width at Half Maximum (FWHM)

In this dataset, stellar point sources were modeled using two-dimensional Gaussian point-spread functions (PSFs), where FWHM served as a metric for image quality and effective spatial resolution. Stellar detections were identified using the DAOStarFinder algorithm [33], applied to sigma-clipped, median-subtracted frames with a 5σ threshold. Each detection underwent localized PSF fitting: a 50×50 px subimage centered on the source was extracted and fit with a two-dimensional Gaussian via a Levenberg–Marquardt least-squares optimizer [34], initialized with FWHM≈3 px. The fit returned standard deviations $\left(\sigma_{x},\sigma_{y}\right)$, which were converted to axis-specific FWHM values using the relation FWHM=2.355σ. An arithmetic mean of the two axes was then computed to yield a scalar FWHM value per source:(1)$${FWHM}_{mean}=\frac{1}{2}\left({FWHM}_{x}+{FWHM}_{y}\right).$$
Outliers with nonphysical fits (0<FWHM>30 px) or contamination from edge effects were excluded from analysis. To assess temporal and spatial stability, FWHM values were aggregated in 10-min bins to compute the mean (μ), standard deviation (σ), and an edge-to-center ratio ρ, defined as: (2)$$\rho =\frac{{\langle FWHM\rangle }_{edge}}{{\langle FWHM\rangle }_{center}}.$$
The “center” region was defined as a 400×400 px window within the 1024×1024 detector array (300<x<700, 300<y<700), chosen to minimize vignetting and boundary artifacts while retaining adequate stellar sampling. These statistical measures—μ, σ, and ρ—provided a compact summary of image quality evolution over time and revealed consistent peripheral PSF broadening (ρ>1), indicative of mild off-axis optical degradation.

### 3.3. Astrometric Residuals

Astrometric accuracy was assessed by comparing the measured field positions of stars with their catalog references in the International Celestial Reference System (ICRS). Each image was plate-solved using [32] to obtain a WCS transformation from detector pixels to celestial coordinates. The field and catalog coordinates, expressed as RA and DEC $\left(\alpha_{field},\delta_{field}\right)$ and $\left(\alpha_{cat},\delta_{cat}\right)$, are given in degrees. Astrometric residuals in right ascension and declination were computed in the tangent-plane projection, with the RA residual scaled by cosδ to represent on-sky angular separations.

### 3.4. Photometric Residuals

Astrometry.net [32] identified 31,759 unique stars in the dataset. The flux values returned in the Astrometry.net output tables are computed by its internal source-extraction algorithm, which estimates and subtracts the local sky background before integrating the object signal. These background-subtracted stellar fluxes were therefore used directly in the photometric calibration. Instrumental magnitudes were derived from the plate-solved measured fluxes [35]: (3)$$m_{inst}=-2.5\;\log_{10}\left(F\right)$$
Catalogue magnitudes in the Johnson *V* band were obtained from Tycho–2 photometry [36] using(4)$$V=V_{T}-0.090\;\left(B_{T}-V_{T}\right)$$

Field-dependent response variations were modeled with a quadratic radial term, $k_{vig}\;r^{2}$, which provides a standard correction for vignetting and flat-field residuals in photometry. A photometric calibration was then performed via(5)$$V_{cat}=ZP+s\;m_{inst}+k_{vig}\;r^{2},$$
where $V_{cat}$ is the catalogue magnitude, $m_{inst}$ is the measured instrumental magnitude, ZP is the photometric zeropoint, *s* is the instrumental-to-catalogue magnitude slope accounting for detector gain nonlinearity, and $k_{vig}$ is the quadratic vignetting coefficient as a function of radial distance *r* from the field center.

#### 3.4.1. RSO Aperture Photometry

Photometric measurements of the annotated RSOs were obtained using circular aperture photometry. A fixed aperture radius and concentric background annulus, with inner and outer radii determined from the FWHM evaluation, were adopted to capture the PSF while minimizing background contamination. Each image was converted to a scalar intensity array to maintain consistent photometric scaling across the dataset. Net source fluxes were calculated by subtracting the background contribution, estimated as the median intensity within the annulus multiplied by the number of aperture pixels, from the total aperture sum. The local background noise was characterized using the median absolute deviation (MAD), scaled to approximate Gaussian statistics (σ≈1.4826 MAD).

Apertures intersecting image boundaries or containing saturated pixels were automatically flagged and excluded from further analysis.

#### 3.4.2. Application of Stellar Calibration

Background-subtracted RSO fluxes were converted to calibrated magnitudes using the photometric parameters ($ZP,\;s,\;k_{vig}$) obtained from the stellar field solution. Each RSO coordinate was matched to the stellar calibration interval covering consecutive frames. Because the stellar zeropoints were derived from summed stacks, the effective zeropoint for a single RSO frame was adjusted as(6)$$ZP^{\prime }=ZP-s\;\left[\;2.5\log_{10}\left(27\right)\;\right]$$
Calibrated magnitudes were then computed as(7)$$V_{RSOemp}=ZP^{\prime }+s\;m_{inst}+k_{vig}\;r^{2}$$

The factor of 27 corresponds to the number of images combined in each stacked frame used for the stellar field solution, requiring an adjustment to match single-exposure RSO frames.

#### 3.4.3. Physics-Based Throughput Model

Absolute photometric estimates of the RSOs were also derived using a physics-based throughput model, which converts measured photoelectron counts into apparent magnitudes [35]. The detected signal in electrons is linearly proportional to the incident photon flux, telescope collecting area, exposure time, detector quantum efficiency, and total optical throughput. The explicit photon-budget formulation was adapted from [31] and expressed as(8)$$V_{RSOphys}=-2.5\;\log_{10}\;\left(\frac{N_{src}}{N_{0,V}\;A\;t_{exp}\;QE\;\eta_{sys}}\right)$$
where $N_{0,V}=10^{10}$ photons m^−2^ s^−1^ represents the photon flux of a zero-magnitude Johnson *V* source, $A=\pi {\left(D_{ap}/2\right)}^{2}$ is the telescope collecting area, $t_{exp}$ is the exposure time, and QE is the detector quantum efficiency. The total system throughput,(9)$$\eta_{sys}=T_{lens}\;T_{win}\;T_{atm}\;F_{ap}$$
combines the lens ($T_{lens}$) and window ($T_{win}$) transmissions, the atmospheric transmission ($T_{atm}$), and the fractional encircled-energy correction ($F_{ap}$), computed for a Gaussian PSF with width given by the FWHM and the adopted aperture radius. The atmospheric transmission using the airmass formulation of Kasten & Young [37],(10)$$T_{atm}=10^{-0.4\;k_{V}\;f_{col}\;{\left[\cos z+0.50572\;{\left(96.07995-z\right)}^{-1.6364}\right]}^{-1}}$$
where *z* is the zenith angle in degrees, $f_{col}=0.0053$ is the fractional atmospheric column above the observer (derived from the pressure ratio at 36.7 km altitude, ≈5 mbar/1013 mbar), and $k_{V}=0.20$ mag airmass^−1^ is the extinction coefficient in the Johnson *V* band. At an observing elevation of 45°, this yields $T_{atm}=0.999$, corresponding to a negligible extinction of $A_{V}≃0.001$ mag.

### 3.5. RSO Angular Rate and Phase Angles

Astrometric calibration was applied to each annotated detection to obtain celestial coordinates and apparent motion. Image-plane measurements were transformed to equatorial coordinates (α,δ) in the ICRS using WCS solutions, and detections sharing a common object identifier were chronologically ordered to estimate apparent angular velocity from successive great-circle separations divided by the corresponding time intervals (${\mathrm{arcsec s}}^{-1}$). Solar ephemerides were computed using a simplified analytical solar position model to obtain $\left(\alpha_{⊙},\delta_{⊙}\right)$. The solar elongation ε (target–Sun separation) and phase angle ϕ=180°−ε were then derived [38]. To ensure consistency with the image astrometry, both the object and solar directions were referenced to the same observer by constructing a topocentric frame from GPS-derived site coordinates (latitude, longitude, Altitude). In this frame, the Sun’s apparent coordinates were obtained from planetary ephemerides, and ϕ≈180°−ε provides an approximation accurate to the arcminute level for Earth-orbiting targets without range information, thereby minimizing parallax-related systematics in ϕ.

### 3.6. TLE-Based Correlation of RSOs

For each object group and its astrometric detections, a correlation was performed against publicly available LEO TLEs from Space-Track.org [39]. Apparent topocentric predictions were generated for every detection at the observation epochs. The GPS telemetry was used for the gondola’s position. Predicted celestial coordinates $\left(\alpha_{pred},\delta_{pred}\right)$ were compared to the measured (α,δ) to form tangent-plane residuals (ΔαcosδandΔδ) (in arcseconds), together with the great-circle separation. To account for correlated errors between right ascension and declination residuals, a covariance-weighted Mahalanobis distance was used as the statistical association metric [40,41]. This approach normalizes each residual vector by the empirical uncertainty structure, providing a scale-invariant measure of consistency between the measured and predicted positions:(11)$$D^{2}\;=\;r^{T}\;S^{-1}\;r,\;r=\left[\begin{array}{c} \Delta \alpha \cos\delta \\ \Delta \delta \end{array}\right]$$

$\chi^{2}$ gate with 2 degrees of freedom was applied at the 99.7% quantile ($D^{2}\leq \chi_{0.997,2}^{2}\approx 11.83$) to flag statistically consistent matches. For each observation, the top five candidates by $D^{2}$ were retained, and the best match (minimum $D^{2}$) was selected to report the NORAD identifier, predicted coordinates, separation, residuals, and Mahalanobis metrics.

### 3.7. RSO Detection

Image stacking was used to combine frames from each sequence into a single composite image. In stacked images, RSOs appear as elongated streaks while stars remain stationary. Stacking improves the photometric signal-to-noise ratio (SNR) and computational efficiency and was applied for stellar photometric residual analysis. However, motion of the observer can degrade stacked image quality; therefore, RSO detection operated on individual frames, while the streak detection used the stacked image sequences. Algorithm performance was previously evaluated on this dataset in [29] using precision, recall, and F1-score metrics, where the F1-score represents the harmonic mean of precision and recall and provides a balanced measure of detection accuracy. The 157-min SPP was critical for RSO detection. A total of 1000 image sequences, comprising 27,027 frames, were processed.

Elongated sources were extracted from plate-solved star-field images [32] after masking catalogued stars to suppress stationary flux. Adaptive thresholding and OpenCV’s findContours function [42] were used to isolate potential streaks, and geometric filters removed spurious detections. Endpoints of valid streaks were used to estimate apparent length, SNR, and celestial coordinates. Of the three frame-differencing methods tested, Adjacent Frame Differencing (AFD), Median Frame Differencing (MFD), and Proximity Frame Thresholding (PFT), the PFT approach [29] achieved the best accuracy and false-positive suppression. PFT combines frame differencing with temporal proximity filtering to link transient detections into continuous tracks, effectively rejecting short-lived artifacts and noise. This approach enables reliable discrimination between RSOs and background stars, forming the basis for subsequent photometric and astrometric analyses.

## 4. Results

### 4.1. Attitude Determination

Attitude was recovered from each frame, yielding Euler angles for an intrinsic XYZ rotation in the ECI frame under intrinsic, active, right-hand-rule conventions. Figure 3 shows the attitude time series over the stabilized pointing period. Because the raw yaw angle exhibits a nearly linear drift of 14.67° $h^{-1}$, consistent with the Earth’s rotation rate of 15° $h^{-1}$, a best-fit linear trend was removed to isolate the short-timescale yaw residuals. These residuals are comparable in amplitude to the fluctuations observed in pitch and roll, confirming that the cumulative yaw drift does not affect short-term pointing stability. Quantitatively, the attitude time series exhibited standard deviations of approximately 0.10°, 0.15°, and 0.91° in yaw residual, pitch, and roll, respectively, indicating stable pointing performance in all axes during the observation period.

### 4.2. Full Width at Half Maximum (FWHM)

Table 2 summarizes image quality across the six observation intervals, each comprising 81 consecutive frames. Using the plate scale of 104 arcsec ${\mathrm{px}}^{-1}$, the mean stellar FWHM values ranged from 377.5 to 404.6 arcsec, with an overall mean of 387.9 arcsec (SD 11.4 arcsec). The smallest FWHM occurred during 04:56–05:26 a.m., and the largest during 05:26–05:33 a.m. Per-interval star counts ranged from 6876 to 11,304 detections (center: 937–2388; edge: 5939–9484). Field uniformity, quantified using the edge-to-center FWHM ratio ρ, ranged from 1.011 to 1.095 (mean 1.058), indicating mild but measurable off-axis broadening in most intervals. These results are visualized in Figure 4.

### 4.3. Astrometric Residuals

Figure 5 presents the two-dimensional distributions of astrometric residuals—differences between field-measured and catalog reference coordinates in right ascension (Δ${\mathrm{RA}}^{*}$) and declination (ΔDec)—for each observation interval. The residual clusters are tightly centered near the origin, indicating high alignment accuracy throughout the observation periods. Across all intervals, the RMS residuals were 67.4 arcsec in right ascension and 81.1 arcsec in declination, corresponding to ∼0.65–0.78 px. These values confirm consistent sub-pixel astrometric performance and are in agreement with the visual concentration seen in the hexbin plots. Residual distributions are nearly circular, showing no significant elongation and implying negligible correlation between Δ${\mathrm{RA}}^{*}$ and ΔDec.

Pearson correlation coefficients are reported in each panel of Figure 5. For the first five intervals, the coefficients lie between −0.12 and +0.07, confirming that the residuals are statistically uncorrelated, while the larger coefficient in the final interval (r=−0.634) reflects the much smaller number of available residuals (n=1815) in that six-minute segment. A slight broadening appears in the final interval (05:26–05:33 a.m.).

### 4.4. Photometric Residuals

Stacked image sequences, each comprising 27 consecutive exposures, produced high-SNR photometric measurements suitable for global calibration. Using all stacked images sequences, the derived relation between instrumental flux and Tycho–2 catalogue magnitudes was(12)V=0.791x+8.245,
with a RMS residual of 0.37 mag (Figure 6). Residuals showed no significant dependence on field radius, and the median offset of −0.036 mag confirmed the absence of measurable vignetting or flat–field bias. The Pearson correlation coefficient between field radius and photometric residuals was r=+0.018, indicating that the residuals are statistically uncorrelated with detector position.

Across all stacked sequences, the calibration parameters were consistent and stable. The mean zeropoint was 〈ZP〉=8.22±0.61mag and the mean slope 〈s〉=0.77±0.13, both in excellent agreement with the global fit parameters (ZP=8.245, s=0.791). The average RMS residual across all stacks was 0.32±0.16mag, and the fitted vignetting coefficient remained below $|k_{vig}|\leq 5\times 10^{-6}$, indicating uniform optical throughput across the detector field. Grouping the data into six observing sessions (02:56–05:33 UTC) revealed no systematic temporal drift, with session-averaged zeropoints ranging from 8.13 to 8.49mag, slopes from 0.75 to 0.83, and RMS values between 0.26 and 0.48mag.

### 4.5. RSO Characterization

A total of 500 RSOs were manually identified and annotated using AstroImageJ v5.5.1.00 [44], which enabled precise pixel-level marking for each detection. From this study, a representative subset of 20 unique RSOs—monitored continuously over a 6-min interval beginning at 04:26 a.m.—was selected for detailed astrometric and photometric characterization. The selection was designed to capture a broad range of apparent brightness, angular velocity, and trajectory orientations across the field of view. Targets were chosen to avoid edge artifacts and ensure full-frame streak coverage, thereby reducing systematic uncertainties in centroiding and flux extraction. While not strictly randomized, the subset was intended to be broadly representative of the RSO population characteristics observed in the wider dataset.

#### 4.5.1. Cross-Comparison of Photometric Estimates

To validate the absolute photometric scale and assess method-dependent systematics, the empirical stellar-calibrated magnitudes ($V_{RSO\;emp}$), derived from the field calibration using $\left(ZP,\;s,\;k_{vig}\right)$, were compared with physics-based throughput magnitudes ($V_{RSO\;phys}$) computed from the photon-budget model incorporating system throughput $\eta_{sys}$, aperture correction $F_{ap}$, and atmospheric transmission $T_{atm}$.

Agreement was evaluated on a paired, per-object basis for 3294 annotations using the magnitude difference(13)$$\Delta m\equiv V_{RSO\;phys}-V_{RSO\;emp},$$
from which the mean bias $\bar{\Delta m}$, standard deviation $\sigma_{\Delta m}$, and Bland–Altman limits of agreement $\bar{\Delta m}\pm 1.96\;\sigma_{\Delta m}$ were computed. A linear regression(14)$$V_{RSO\;phys}=a+b\;V_{RSO\;emp},$$
tested for residual nonlinearity (b≠1) or offset (a≠0). To quantify both association and agreement between the two magnitude estimates, we computed Pearson’s correlation coefficient (*r*), Spearman’s rank correlation coefficient (ρ), and the concordance correlation coefficient (CCC). Pearson’s *r* measures linear correlation, Spearman’s ρ captures monotonic (but potentially non-linear) relationships, and the CCC assesses how closely the paired magnitudes lie along the 1:1 line, i.e., their absolute agreement in both scale and offset.

The two magnitude systems exhibit distinct absolute scales, with mean values $\langle V_{\mathrm{RSO}\;\mathrm{emp}}\rangle =-1.54\pm 0.57$ mag and $\langle V_{\mathrm{RSO}\;\mathrm{phys}}\rangle =3.91\pm 0.69$ mag. The mean paired difference $\bar{\Delta m}=5.45$ mag corresponds to a constant scale bias between the calibrations, while the dispersion of 0.58 mag yields Bland–Altman limits of [+4.31,+6.58] mag C The correlation coefficients were CCC = 0.02, Pearson r=0.60, and Spearman ρ=0.69. The Bland–Altman diagram in Figure 7 shows a dense, nearly symmetric core of points within the 95% limits of agreement, together with broader “wings” that extend beyond these limits. The central cluster is consistent with random scatter around the mean bias of +5.45 mag.

#### 4.5.2. Apparent Angular Velocity of RSOs

The apparent angular velocities of the 20 annotated RSOs were computed from successive astrometric detections within each tracklet by dividing the great-circle separations by their corresponding time intervals. The mean angular rates obtained from this procedure are those summarized in Table 3. The resulting mean angular velocities span $7.3\times 10^{2}$–$8.4\times 10^{3}$ arcsec $s^{-1}$ (0.20–$2.33{}^{\circ}\;s^{-1}$), with a median of $1.6\times 10^{3}$ arcsec $s^{-1}$ and an interquartile range of 1.1–$2.3\times 10^{3}$ arcsec $s^{-1}$. The overall mean, $2.1\times 10^{3}\pm 1.7\times 10^{3}$ arcsec $s^{-1}$, is consistent with near-Earth objects observed at slant ranges of a few hundred to a few thousand kilometers, under the kinematic bound $\omega \approx v_{⊥}/R\leq 7.7\;$ km s^−1^/*R*, where $v_{⊥}$ is the transverse velocity component and *R* the slant range. Within-track standard deviations ranged from 200 to 5100 arcsec $s^{-1}$.

#### 4.5.3. Phase-Angle Distribution and Illumination Geometry

Phase angles (ϕ) were derived for 20 annotated RSOs using the topocentric geometry, as tabulated in Table 3. The corresponding solar elongations (ε) ranged from 133.5° to 167.2°, yielding phase angles ϕ=12.8°–46.5° with a mean of 29.8° ± 7.8° and an interquartile range of 24.1°–36.1°. Phase angles between 25° and 35° dominated the sample, and per-object statistics showed tight clustering (Δϕ<1°). All observations occurred at large elongations near the anti-solar direction, consistent with nighttime passive imaging geometry.

### 4.6. TLE-Based Correlation of RSOs

For each observation frame, catalog candidates were ranked by astrometric consistency using the Mahalanobis distance $D^{2}$. The top-ranked candidate in each frame was considered the frame-level match. Across all frames within an RSO sequence, the final identity was assigned to the catalog object (NORAD ID) that achieved the majority of rank-1 selections. In cases of ties, precedence was given to smaller average rank, lower mean $D^{2}$, and greater number of appearances. For each sequence, we report the number and fraction of rank-1 wins, the average rank, and the mean $D^{2}$, which together quantify the stability of the identification.

Table 3 summarizes the best catalog correlations obtained for each detection sequence, together with the mean Mahalanobis distance squared ($\bar{D^{2}}$) as a quantitative measure of astrometric consistency. The identified objects span a range of orbital classes, dominated by small debris fragments such as THORAD AGENA D DEB and FENGYUN 1C DEB, and a smaller subset of intact payloads including COSMOS 2118, NINGXIA-12, and YAOGAN 14. Most detections exhibit $\bar{D^{2}}$ values below 0.05, corresponding to Mahalanobis distances of less than 0.25 σ when normalized by the empirical residual covariance. These low values indicate close agreement between the measured and predicted topocentric positions, affirming the precision of the astrometric calibration and the reliability of the catalog correlations.

Apparent angular velocities for the correlated objects cluster between approximately 700 and 3000 arcsec $s^{-1}$, consistent with low-Earth-orbit trajectories observed at moderate phase angles. A few detections display higher mean rates (>4000 arcsec $s^{-1}$), which likely correspond to lower-altitude debris or rapidly transiting fragments. The overall distribution confirms that the covariance-weighted matching and majority-rank aggregation effectively suppress frame-level outliers while maintaining sub-arcminute astrometric coherence across sequences.

### 4.7. RSO Detection

Representative examples of single-frame and stacked detections are shown in Figure 8, illustrating both isolated and overlapping RSO streaks within the wide-field images. The detection performance of the combined streak-detection and Proximity Filtering and Tracking (PFT) algorithms is summarized in Table 4.

A total of 387 RSO streaks were identified throughout the observation period. The number of detected streaks varied across the observation period. The lowest count occurred between 3:26 a.m. and 3:56 a.m., corresponding to the darkest portion of the night. As the observations approached astronomical dawn, both the number and mean angular length of the detected streaks increased, peaking at 4679 arcsec between 4:56 a.m. and 5:26 a.m. The corresponding PFT detections followed the same temporal trend, with the highest detection counts occurring during this interval. Near dawn, the mean signal-to-background ratio (SNR) began to decline, consistent with the gradual brightening of the sky.

## 5. Discussion

### 5.1. Attitude Determination

After detrending, the yaw residuals behaved similarly to the pitch and roll fluctuations, indicating that pointing jitter was dominated by short-timescale gondola dynamics rather than systematic drift. The near-constant pitch reflects a stable elevation relative to the celestial equator, supporting uniform star density and consistent WCS solutions throughout the sequence. The larger roll excursions are consistent with residual gondola dynamics and stabilization corrections (e.g., wind-induced motion or control updates) and explain the slight broadening observed in the astrometric residuals near sunrise. These behaviors motivate the use of rolling residual covariances in the correlation pipeline to capture time-varying pointing jitter.

Dedicated fine-pointing balloon systems, such as NASA’s Wallops Arcsecond Pointer (WASP) [45] and the SuperBIT telescope [46], achieve arcsecond-level line-of-sight stability and are specifically engineered for precision optical tracking. In contrast, RSONAR was flown on a CNES/CSA STRATO Science gondola, whose attitude control system is designed for coarse pointing at the sub-degree level rather than fine stabilization. The attitude estimates used in this work were derived from 27-frame stacked images, which provide robust star solutions but inherently smooth short-timescale motion. Using single-frame attitude solutions would yield more precise instantaneous pointing estimates, but doing so would increase the computational load by roughly a factor of 26 and was therefore not practical for the present analysis.

### 5.2. FWHM

The consistent mean FWHM and narrow confidence intervals demonstrate that optical performance and focus stability were maintained throughout the stabilized pointing period. The mild peripheral broadening (ρ≈1.06) likely reflects a combination of optical field curvature and minor attitude jitter rather than defocus drift. The absence of significant FWHM growth over time confirms that temperature variations at float altitude did not meaningfully degrade focus or image sharpness. These results validate the payload’s optical design and confirm that the measured image quality was sufficient to support precise centroiding and astrometric plate solving across the field of view.

### 5.3. Astrometric Residuals

The sub-pixel residuals confirm that the astrometric calibration and WCS solutions remained geometrically stable throughout the flight. The minor broadening during the final interval coincides with increased gondola motion near sunrise, suggesting small pointing perturbations rather than calibration drift. Because the final interval spans only six minutes, far fewer frames and stellar detections were available, which reduces the number of astrometric residuals and contributes to the different hexbin appearance in that panel. The slight elongation of the residual cloud is consistent with the increased gondola motion near sunrise and does not indicate a change in the underlying astrometric solution. This interpretation is supported by the negligible Pearson correlations in the first five intervals, which demonstrate that RA and Dec residuals are not systematically coupled and that the astrometric calibration remained unbiased. The residual symmetry indicates that systematic distortions, such as differential rotation or skew between RA and Dec, were effectively mitigated by the plate-solving process. The slight increase in dispersion toward the field edges is consistent with the expected behavior of a wide-field optical system, where geometric distortion increases off-axis. Overall, these results validate the robustness of the astrometric registration pipeline and demonstrate that the RSONAR payload maintained unbiased alignment across all observation intervals.

### 5.4. Photometric Residuals

The tight linear relation between instrumental flux and catalogue magnitude, along with the small RMS scatter, demonstrates that the photometric response of the imager was both linear and temporally stable. The negligible dependence of residuals on field radius and the extremely low vignetting coefficient confirm that the optical train and detector maintained uniform throughput across the full field of view. The lack of systematic variation in zeropoint or slope among the six observing sessions indicates that atmospheric transmission above 36 km altitude was effectively constant and that no measurable degradation in detector sensitivity occurred during the flight. These results support the use of a single global calibration for both stellar and RSO photometry.

A small secondary cloud of points below the main flux–magnitude relation (Figure 6) appears in a limited number of stacked sequences. These points are associated with brief intervals of increased gondola motion, which reduce the recovered stellar flux in the stacks and cause some faint stars to fall below the primary calibration trend. The same mechanism explains the “wings” observed in the Bland–Altman comparison. During intervals of elevated gondola motion, PSF broadening and slight frame-to-frame misregistration decrease the stellar flux used in the calibration transfer. Because the stellar calibration is applied directly to single-frame RSO photometry, these flux underestimates manifest as larger physics–empirical magnitude differences. Such discrepancies are therefore observational and processing-related rather than intrinsic limitations of the reflectance model. Future analyses could mitigate these effects through motion-aware stacking, PSF-shape diagnostics, and per-image weighting of calibration frames.

### 5.5. RSO Characterization

The cross-comparison confirms that the empirical and physics-based photometric methods differ by a constant offset of approximately 5.45 mag, attributable to their independent zeropoint definitions. Despite the absolute bias, the narrow scatter ($\sigma_{\Delta m}=0.58$ mag) and moderate correlations (r=0.60, ρ=0.69) demonstrate that both methods preserve consistent relative photometric behaviour. This indicates that either calibration can be used interchangeably for relative analyses, such as light-curve extraction or brightness ranking within the field. The low dispersion also validates the internal consistency of the empirical calibration derived from stellar references, reinforcing its applicability to RSO photometry under the stable illumination and optical conditions present during the flight. Overall, these results demonstrate that the dual-purpose star tracker can provide reliable brightness estimates for resident space objects even when absolute flux calibration is uncertain.

The derived angular velocity distribution is characteristic of low-Earth-orbit objects observed near the local terminator. The spread in measured rates primarily reflects projection effects as the line of sight evolves across the sky, producing higher apparent motion for objects with large transverse components or shorter instantaneous ranges, and lower values when the motion is more radial or the object more distant. The absence of slow-moving (<500 arcsec $s^{-1}$) or stationary signatures confirms that the detections exclude medium- and geostationary-Earth orbits. Although ω provides a strong geometric constraint, it does not uniquely determine altitude or inclination; multiple orbital configurations can yield the same apparent rate. Recovering complete orbital parameters would require additional observables such as curvature, acceleration, or parallax from multi-site observations. These results therefore demonstrate the consistency of the measured angular velocities with LEO dynamics while highlighting the geometric degeneracies inherent in single-station, short-arc observations.

The dominance of moderate phase angles (25°–35°) indicates that all RSOs were observed under stable, partially backscattered illumination typical of near-terminator geometries. Such conditions maximize the illuminated cross-section of sunlit objects while maintaining high contrast against the dark sky, enhancing detectability in wide-field imaging. The narrow range of ϕ across all targets confirms that illumination remained effectively constant during the observation sequence, ensuring photometric comparability between RSOs. This mid-ϕ regime aligns with mixed diffuse–specular scattering models for satellite surfaces [38,47], which predict only gradual brightness attenuation with increasing phase angle, retaining over 80% of the opposition brightness by ϕ≈57° [48]. Consequently, the dataset represents a well-defined, sunlit illumination geometry ideally suited for evaluating RSO photometric and astrometric performance in SSA applications.

### 5.6. TLE-Based Correlation of RSOs

While the Mahalanobis-based majority correlation framework yields internally consistent astrometric matches, it remains limited by several methodological and operational factors. The approach is fundamentally a local, angles-only decision rule—it does not enforce dynamic consistency across time or orbital arcs and therefore cannot resolve ambiguities in regions where RSO trajectories intersect within the focal plane. In such conjunction scenarios, majority-rank logic alone cannot distinguish between objects with overlapping apparent motion, requiring temporal segmentation and pre-conjunction matching.

Additional sources of uncertainty arise from the dataset itself. The payload’s onboard GPS failure necessitated reconstructing frame timestamps from the instrument’s internal clock, later aligned to the gondola GPS time via offset–drift correction. This process introduced residual timing uncertainty on the order of ±2–3 s. At typical low-Earth-orbit angular rates (1000–3000 arcsec $s^{-1}$), this translates to positional errors of several thousand arcseconds, effectively precluding precise orbit determination from this dataset. Consequently, the analysis was restricted to statistical correlation and cross-identification using apparent topocentric positions, rather than full orbital reconstruction.

Furthermore, the RSOs observed during this flight constitute short-arc detections—each track spans only a small fraction of a full orbit. When combined with single-station, angular-only measurements, such short arcs lack sufficient geometric leverage for unique OD or multi-pass orbit linkage. The single wide-field imager configuration, without cross-sensor parallax, further constrains the analysis to purely angular correlations.

### 5.7. RSO Detection

The temporal variation in streak and PFT detections demonstrates that RSO visibility is strongly governed by illumination geometry and background brightness. The increase in detections near astronomical dawn reflects the improved reflectivity of sunlit RSOs as the Sun approached the local horizon, while the subsequent SNR decline marks the onset of sky-background saturation. These results identify the period around astronomical twilight as the optimal window for stratospheric RSO observation, offering balanced illumination and favorable phase angles between the Sun, object, and observer. The consistent correspondence between streak counts and PFT detections confirms the robustness of the PFT method, which has previously shown superior detection accuracy and false-positive suppression compared to other differencing techniques [29]. The findings also emphasize the need for adaptive thresholding to maintain detection sensitivity under rapidly changing light conditions. Overall, the detection analysis verifies the secondary functionality of the dual-purpose star tracker as an effective RSO detection instrument for SSA operations.

### 5.8. Limitation and Future Work

Future stratospheric SSA campaigns should incorporate GPS-synchronized timing systems, and multi-site observations to enable admissible-region initialization and batch orbit-determination refinement consistent with modern Uncorrelated Track frameworks. Expanding to multi-site operations with varied imagers across stratospheric and ground-based platforms will provide cross-validation and highlight the advantages of the stratospheric vantage point. Although the present study confirms the feasibility of astrometric correlation and statistical identification from a single wide-field sensor, its short observational arcs, lack of parallax geometry, and residual timing uncertainty from the onboard GPS failure limited the analysis to angles-only association rather than full orbit determination. Implementing these enhancements will establish stratospheric balloon missions as recoverable, low-cost testbeds for SSA, supporting light-curve extraction, improved catalog cross-referencing, and dynamic RSO correlation that bridges ground and space-based architectures.

## 6. Conclusions

In conclusion, the dual-purpose functionality of a commercial ST for SSA was successfully demonstrated using imagery acquired during the 2022 RSONAR stratospheric flight. The low-cost, commercial off-the-shelf ST effectively performed both attitude determination and RSO detection tasks, confirming its viability as a compact SSA sensor. Image quality remained stable throughout the flight, with consistent FWHM and minimal field distortion, supporting sub-pixel astrometric accuracy and robust plate solutions. Photometric calibration produced linear, field-uniform results, enabling reliable brightness estimation for both stellar and RSO targets.

A total of over 500 RSOs were visually confirmed, and 20 representative sequences were analyzed in detail to characterize their astrometric, photometric, and dynamic properties. Apparent angular velocities (700–3000 arcsec $s^{-1})$ and phase-angle distributions (25–35°) were consistent with sunlit LEO objects observed near the anti-solar direction. Using a covariance-weighted Mahalanobis distance framework, catalog correlation against TLEs achieved sub-arcminute agreement, validating the reliability of topocentric association from a single wide-field imager.

Overall, these results mark the transition from a dual-purpose ST concept to a validated dual-use ST, demonstrating that commercial star trackers can fulfill both attitude determination and passive SSA functions within a unified, low-cost optical system. This advancement broadens their role beyond spacecraft navigation to active space domain awareness.

## Figures and Tables

**Figure 1 sensors-26-00179-f001:**
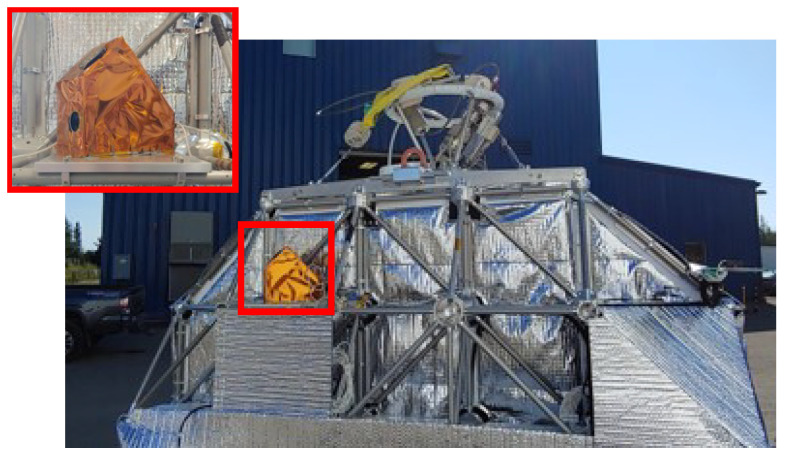
RSONAR payload integrated on the CSA gondola prior to flight (highlighted in red). Inset: close-up view of the RSONAR payload.

**Figure 2 sensors-26-00179-f002:**
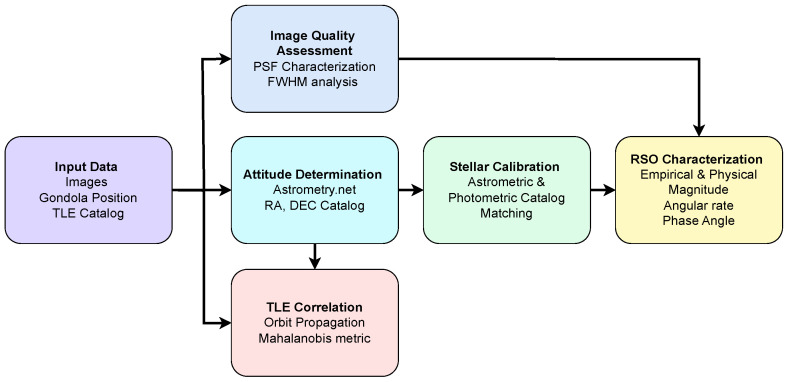
Overview of the processing methodology, including image acquisition, attitude determination, image-quality assessment, stellar calibration, RSO photometric and astrometric analysis, and final TLE-based correlation.

**Figure 3 sensors-26-00179-f003:**
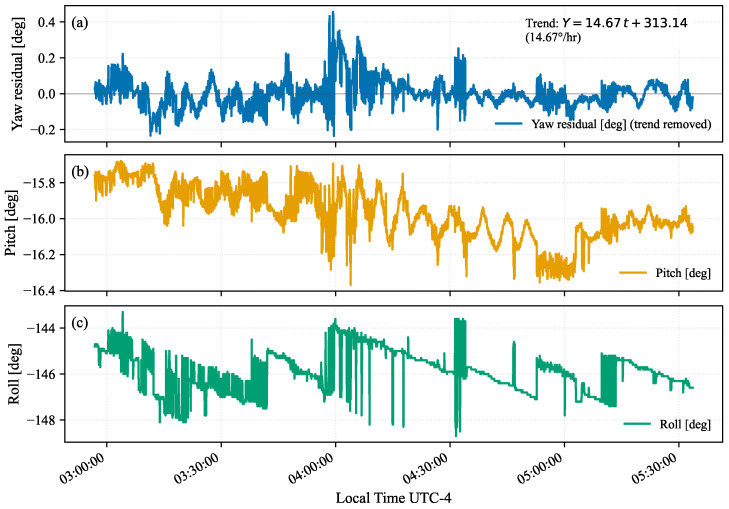
Attitude time series during the stabilized pointing period [43]. Yaw residuals (blue) in (**a**) are shown after removing a 14.67° $h^{-1}$ drift. Panels (**b**,**c**) show pitch (yellow) and roll (green). Residual amplitudes indicate stable short-timescale pointing.

**Figure 4 sensors-26-00179-f004:**
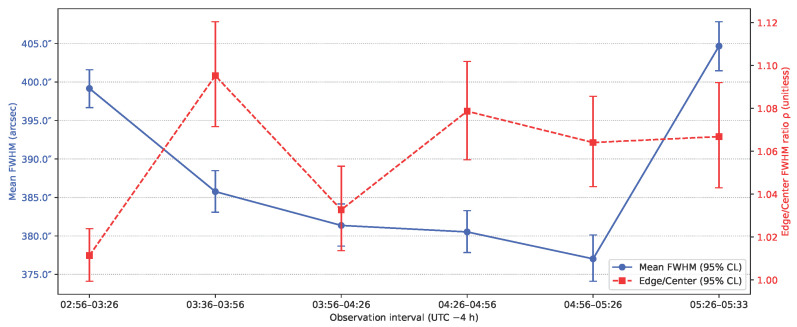
Temporal behavior of image quality and field uniformity. Points show mean stellar FWHM (left axis) and edge-to-center ratio ρ (right axis) per observation interval, with 95% confidence limits.

**Figure 5 sensors-26-00179-f005:**
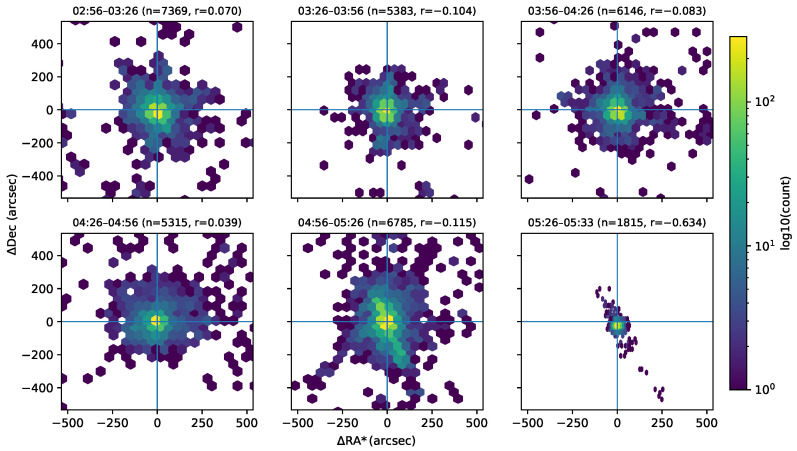
Two-dimensional distributions of astrometric residuals (Δ${\mathrm{RA}}^{*}$, ΔDec) for each observation interval. Each hexbin panel shares the same angular scale (e.g., ±500 arcsec) and color normalization. Color indicates $\log_{10}$ (counts). The residuals remain centered near zero, confirming sub-pixel alignment accuracy across the flight.

**Figure 6 sensors-26-00179-f006:**
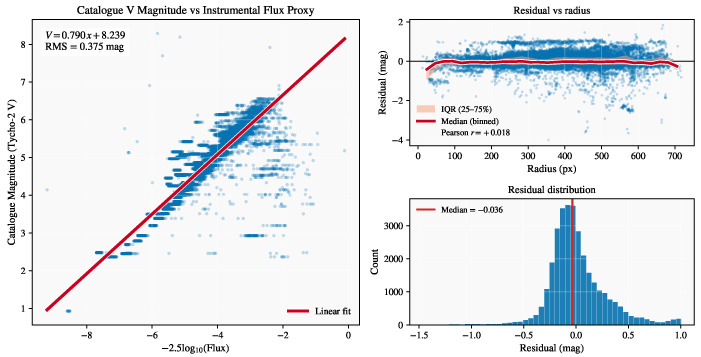
Global photometric calibration diagnostics derived from all stacked image sequences. (**Left**): Tycho–2 catalogue *V* magnitude versus instrumental flux proxy with best-fit linear relation and RMS scatter. (**Top right**): residuals versus field radius with binned median and interquartile range (IQR) with pearson correlation coefficient. (**Bottom right**): residual distribution with median offset indicated.

**Figure 7 sensors-26-00179-f007:**
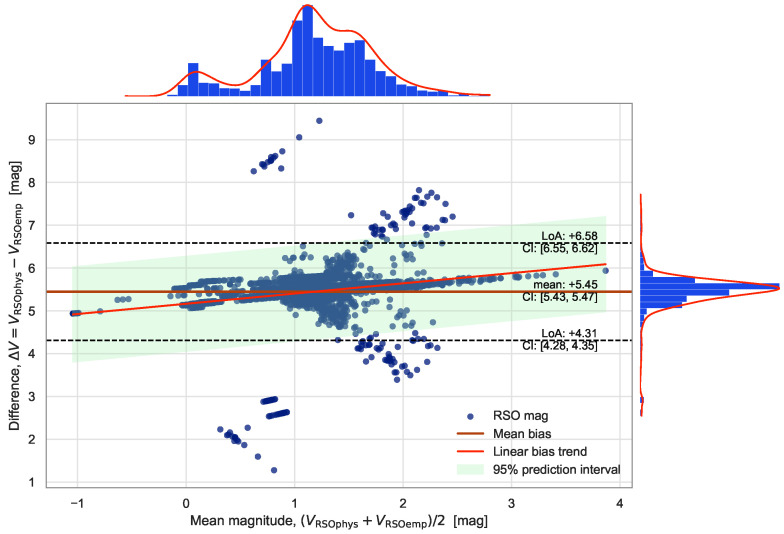
Bland–Altman comparison between the physics-based RSO magnitudes ($V_{\mathrm{RSO}\;\mathrm{phys}}$) and the empirical stellar-calibrated magnitudes ($V_{\mathrm{RSO}\;\mathrm{emp}}$). The solid line marks the mean difference (bias) of $\bar{\Delta m}=+5.45$ mag, and the dashed lines show the 95% limits of agreement (LoA), defined as $\bar{\Delta m}\pm 1.96\;\sigma =\left[+4.31,+6.58\right]$ mag. Shaded regions indicate the 95% confidence intervals (CI) for the bias and the LoA. The compact central cluster demonstrates good relative agreement between the two photometric estimates, while the extended wings correspond to higher-uncertainty measurements.

**Figure 8 sensors-26-00179-f008:**
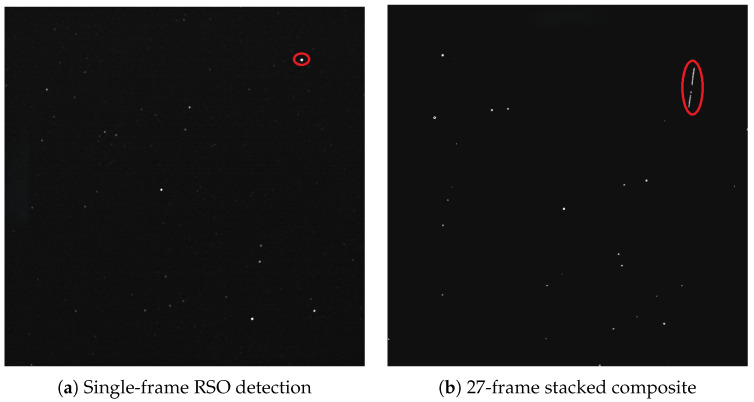
Examples of RSO detections from the RSONAR payload. (**a**) A single 0.1 s exposure acquired at 05:04:31.758 UTC, centered at (RA, Dec) = (344.365°, 16.051°) with a 29.7° × 29.7° field of view. The short linear streaks (circled) correspond to transiting RSOs captured within a single frame. (**b**) A 27-frame stack spanning 05:04:31–05:04:37 UTC, centered at (344.385°, 16.069°) with the same field of view. Stacking increases SNR and reveals fainter objects, while producing longer RSO trails due to object motion over the stacking interval. The stacked image is thresholded for visual clarity.

**Table 1 sensors-26-00179-t001:** Optical, imaging, and detector characteristics of the RSONAR payload during the 2022 STRATOS flight.

Parameter	Value
Aperture diameter ($D_{ap})$	12.5 mm
Focal length	25 mm
Focal Ratio	f/2
Field of View	29.7° × 29.7°
Lens transmission ($T_{lens}$)	0.94
Window transmission ($T_{win}$)	0.98
Pixel size	6.5 μm × 6.5 μm
Pixel scale	104 arcsec/pixel
Chromaticity	Monochrome (unfiltered)
Bit depth	16 bits
Exposure time	100 ms
Effective quantum efficiency ($QE_{eff}$ @ 550 nm)	78%
Dark current ($N_{d,e,s}$ @ −1.5°C)	1.12 e^−^/s/pixel
Read noise ($RN_{e}$)	2.1 e^−^ rms/pixel/read

**Table 2 sensors-26-00179-t002:** Image quality and field-uniformity metrics across six observation intervals. FWHM values are reported in arcseconds; ρ is the edge-to-center FWHM ratio.

Interval (UTC)	Mean FWHM (arcsec)	CL_95_ (arcsec)	ρ
02:56–03:26	399.2	[396.7, 401.6]	1.01
03:36–03:56	385.8	[383.1, 388.5]	1.10
03:56–04:26	381.4	[378.7, 384.2]	1.03
04:26–04:56	380.5	[377.9, 383.3]	1.08
04:56–05:26	377.0	[374.1, 380.1]	1.06
05:26–05:33	404.7	[401.5, 407.8]	1.07
Overall Mean	388.1	—	1.058

**Table 3 sensors-26-00179-t003:** Catalog-correlated RSOs identified through astrometric association. Each entry lists the matched NORAD identifier, catalog designation, mean Mahalanobis distance squared ($\bar{D^{2}}$), the mean angular velocity 〈ω〉 (arcsec $s^{-1}$), and the mean solar phase angle 〈ϕ〉 (deg) over the corresponding observation sequence. Lower $\bar{D^{2}}$ values indicate stronger astrometric agreement with predicted ephemerides.

NORAD	Name	Mean $\bar{D^{2}}$	〈ω〉 (arcsec s^−1^)	〈ϕ〉 (deg)
32153	FENGYUN 1C DEB	0.003	1559.9	36.20
06020	SL-8 R/B	0.025	3686.8	34.11
21032	COSMOS 2118	0.014	8440.1	34.34
42440	NIMBUS 2 DEB	0.025	1396.6	32.86
39818	COSMOS 1823 DEB	0.035	2300.6	38.19
01549	SL-8 DEB	0.035	2499.6	34.83
05507	THORAD AGENA D DEB	0.003	1114.8	22.19
44780	NINGXIA-12	0.013	1543.1	24.53
38257	YAOGAN 14	0.004	728.6	12.74
08296	THORAD DELTA 1 DEB	0.033	3014.6	41.54
34610	COSMOS 2251 DEB	0.043	1900.9	33.34
19786	COSMOS 1995	0.009	2063.8	28.74
21032	COSMOS 2118	0.009	1279.0	26.22
21269	DELTA 1 DEB	0.006	1188.0	26.75
04968	THORAD AGENA D DEB	0.015	1027.9	25.19
04823	COSMOS 374 DEB	0.011	921.5	24.63
30200	FENGYUN 1C DEB	0.016	1019.1	28.21
23859	MSX DEB	0.023	1894.4	35.23
31133	AEROCUBE 2	0.025	2246.4	41.67
22601	SL-16 DEB	0.016	1723.1	27.70

**Table 4 sensors-26-00179-t004:** Summary of RSO streak and PFT detections across observation intervals.

Observation Period	Number of Streaks	Mean Streak Length (arcsec)	Mean SNR	PFT Detections
2:56–3:26 a.m.	65	2036.47	18.51	527
3:26–3:56 a.m.	6	2981.90	18.74	440
3:56–4:26 a.m.	76	3413.58	18.40	3500
4:26–4:56 a.m.	110	4215.95	18.76	7849
4:56–5:26 a.m.	86	4679.19	17.44	7260
5:26–5:33 a.m.	44	4265.96	17.57	2460
Total	387	3781.81	18.22	22,036

## Data Availability

The data presented in this study are available upon request from the corresponding author.

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
