# Peer review of "RSONAR: Data-Driven Evaluation of Dual-Use Star Tracker for Stratospheric Space Situational Awareness (SSA)"

_sensors, 2025, doi:10.3390/s26010179_

Round 1

Reviewer 1 Report

Comments and Suggestions for Authors

L195. Please include a footnote with the hyperlink to astronomy.net.

L287. Even though D_ap can be inferred from the context, it has not been explicitly defined earlier.

L336. Please explicitly define the term F1-score metrics and provide a contextual explanation within the same paragraph.

L362. The entire paragraph is difficult to read and could be replaced with a table accompanied by a short description.

L382. The individual hexbin sizes in the bottom/right panel are slightly different from those in the other panels. Is there a reason for this? The number of residuals is significantly lower and corresponds to only six minutes of stabilised pointing at dawn. Could you please elaborate on this (perhaps within Section 6.7)?

L427. The abbreviation CCC (concordance correlation coefficient) appears in the paragraph without explanation. Please briefly describe why CCC, as well as the Spearman and Pearson coefficients, are required in this context.

L439. Are these RSOs the same as those listed in Table 2? If so, Table 2 only shows 19 objects. See also L620, where it is mentioned that angular velocities and phase-angle distributions were also calculated for the reference RSOs. Could this particular dataset be included in Table 2 as well?

L486. Isn’t the overall yaw variation consistent with the Earth’s rotation (15°/hr) during stabilised pointing extending for about 157 minutes? If not, discuss in more detail why the cumulative yaw increment would not affect the pointing capabilities of the instrument.

Fig. 3. It is not clear from the plot due to the large y-range—are the yaw error values comparable to those of pitch and roll? (e.g., after subtracting the linear fit from the raw yaw data).

Fig. 7. This figure is of particular concern, as it shows highly dispersed—yet apparently symmetrical—data, roughly consistent with an overall linear fit. Could you discuss in greater detail the nature of the “wings” exceeding the limits of agreement, and what could be improved in future analyses of physics-based versus stellar RSO magnitude differences (e.g., in Section 5.8)?

Fig. 7. Please also define the terms limits of agreement (LoA) and CI within the figure caption, not only inside the plot.

Fig. 18. If the figure shows single (left) and stacked (right) RSOs, why are the same objects not observed at the same positions in the stacked image? Do they correspond to different fields of view, observation epochs, or other factors?

Fig. 18. Are the streaks in the left panel the so-called “transiting RSOs”? If so, could you draw boxes around them for illustration, as done in Kunalakantha et al. (2023)? Please also expand the figure caption to include more details, such as the field of view size.

L504 / Sect. 5.3. The pointing capabilities of the instrument appear to be adequate, but how do they compare to those of other missions or expeditions (see Figures 3 & 4)? Please provide additional literature references addressing this issue.

Author Response

We would like to thank the reviewers for their careful reading and constructive comments. All suggested changes have been implemented in the revised manuscript, and highlighted text indicates the edits made in response to the reviews.

Below we address each comment point-by-point.

Comment 1: L195. Please include a footnote with the hyperlink to astrometry.net.

Response: Thank you for the suggestion. A footnote containing the full hyperlink to astrometry.net has been added on page 6 at the first mention of the service

Comment 2: L287. Even though can be inferred from the context, it has not been explicitly defined earlier.

Response: We agree. is now explicitly defined in Table 1 as the aperture diameter.

Comment 3: L336. Please explicitly define the term F1-score metrics and provide a contextual explanation within the same paragraph.

Response: Thank you for this helpful comment. The paragraph has been revised to include a clear definition of the F1-score as the harmonic mean of precision and recall, along with a brief explanation of why it provides a balanced measure of detection performance (line 350).

Comment 4: L362. The entire paragraph is difficult to read and could be replaced with a table accompanied by a short description.

Response: We agree that paragraph is dense. The paragraph has been replaced by a concise summary table (Table 2) reporting per-interval mean FWHM values, 95% confidence limits, and field-uniformity ratios. A shorter explanatory paragraph accompanies the table (beginning at line 378).

Comment 5: L382. The individual hexbin sizes in the bottom/right panel are slightly different from those in the other panels. Is there a reason for this? The number of residuals is significantly lower and corresponds to only six minutes of stabilised pointing at dawn. Could you please elaborate on this (perhaps within Section 6.7)?

Response: All hexbin residual maps use the same grid settings. The difference in appearance originates from the final interval (05:26–05:33), which spans only six minutes. This shorter duration results in substantially fewer detections, leading to lower occupancy in several hexbins and a visually different distribution.

A clarification has been added to Section 5.3 (Astrometric Residuals – Discussion) (line 545):

“Because the final interval spans only six minutes, far fewer frames and stellar detections were available, which reduces the number of astrometric residuals and contributes to the different hexbin appearance in that panel…”

This text also explains that the slight elongation is consistent with increased gondola motion near sunrise and is not indicative of degraded astrometric performance.

Comment 6: The abbreviation CCC (concordance correlation coefficient) appears in the paragraph without explanation. Please briefly describe why CCC, as well as the Spearman and Pearson coefficients, are required in this context.

Response: A sentence has been added in Section 4.5.1 (line 432) explaining why all three metrics are used:

“Pearson’s measures linear correlation, Spearman’s captures monotonic (possibly non-linear) relationships, and the concordance correlation coefficient (CCC) quantifies absolute agreement relative to the 1:1 line.”

Comment 7: L439. Are these RSOs the same as those listed in Table 2? If so, Table 2 only shows 19 objects. See also L620, where it is mentioned that angular velocities and phase-angle distributions were also calculated for the reference RSOs. Could this particular dataset be included in Table 2 as well?

Response: Thank you for noting this discrepancy. One RSO was missing from the previous version of Table 2; all 20 RSOs are now included. In addition, we expanded the table to report the mean angular rate and mean phase angle for each object.

Comment 8: L486. Isn’t the overall yaw variation consistent with the Earth’s rotation (15°/hr) during stabilised pointing extending for about 157 minutes? If not, discuss in more detail why the cumulative yaw increment would not affect the pointing capabilities of the instrument. Fig. 3. It is not clear from the plot due to the large y-range—are the yaw error values comparable to those of pitch and roll? (e.g., after subtracting the linear fit from the raw yaw data).

Response: We appreciate this insightful comment. We quantified the yaw drift by fitting a linear model to the raw yaw series, obtaining a drift rate of 14.67°/hr, consistent with the Earth’s apparent rotation rate of 15°/hr.

To isolate short-timescale motion, this linear trend was removed, and the yaw residuals were compared against pitch and roll. The updated Figure 3 shows detrended yaw residuals, pitch and roll fluctuations, the fitted yaw drift rate annotated in the top panel.

Corresponding text has been added to Section 4.1 (line 491)

“Because the raw yaw angle exhibits a nearly linear drift of 14.67° hr−1, consistent with the Earth’s rotation rate of 15° hr−1, a best-fit linear trend was removed to isolate the short-timescale yaw residuals. These residuals are comparable in amplitude to the fluctuations observed in pitch and roll, confirming that the cumulative yaw drift does not affect short-term pointing stability.”

 And Section 5.1 (line 513):

“After detrending, the yaw residuals behaved similarly to the pitch and roll fluctuations, indicating that pointing jitter was dominated by short-timescale gondola dynamics rather than systematic drift.”

Comment 9: Fig. 7. This figure is of particular concern, as it shows highly dispersed—yet apparently symmetrical—data, roughly consistent with an overall linear fit. Could you discuss in greater detail the nature of the “wings” exceeding the limits of agreement, and what could be improved in future analyses of physics-based versus stellar RSO magnitude differences (e.g., in Section 5.8)?

Response: A detailed explanation has been added to Section 4.5.1 (line 450)

“The Bland–Altman diagram in Fig. 7 shows a dense, nearly symmetric core of points within the 95% limits of agreement, together with broader “wings” that extend beyond these limits. The central cluster is consistent with random scatter around the mean bias of +5.45 mag.”

and expanded in Section 5.4 (line 572):

“The same mechanism explains the “wings” observed in the Bland–Altman comparison. During intervals of elevated gondola motion, PSF broadening and slight frame-to-frame misregistration decrease the stellar flux used in the calibration transfer. Because the stellar calibration is applied directly to single-frame RSO photometry, these flux underestimates manifest as larger physics–empirical magnitude differences. Such discrepancies are therefore observational and processing-related rather than intrinsic limitations of the reflectance model. Future analyses could mitigate these effects through motion-aware stacking, PSF-diagnostics, and per-image weighting of calibration frames.”

Comment 10: Please also define the terms limits of agreement (LoA) and CI within the figure caption, not only inside the plot.

Response: Thank you for your feedback, The caption of Fig. 7 now explicitly defines LoA and CI.

Comment 11: If the figure shows single (left) and stacked (right) RSOs, why are the same objects not observed at the same positions in the stacked image? Do they correspond to different fields of view, observation epochs, or other factors?

Response: The two panels now show updated imagery and correspond to different observation epochs separated by several seconds: Single-frame image: 05:04:31.758 UTC, 27-frame stacked composite: 05:04:31–05:04:37 UTC. Both panels share the same FOV (29.7° × 29.7°) centered at approximately RA = 344.37°, Dec = +16.06°. The caption now states the difference in observing epochs explicitly.

Comment 12: Are the streaks in the left panel the so-called “transiting RSOs”? If so, could you draw boxes around them for illustration, as done in Kunalakantha et al. (2023)? Please also expand the figure caption to include more details, such as the field of view size.

Response: Bounding circles have been added around the transiting RSOs for clarity. The caption now includes RA/Dec of each panel field-of-view size and observation time and identification of the circled RSO streaks

Comment 13: The pointing capabilities of the instrument appear to be adequate, but how do they compare to those of other missions or expeditions (see Figures 3 & 4)? Please provide additional literature references addressing this issue.

Response:  A comparative paragraph has been added to Section 5.1 (line 522):

“Dedicated fine-pointing balloon systems, such as NASA’s Wallops Arcsecond Pointer (WASP) [45] and the SuperBIT telescope [ 46], achieve arcsecond-level line-of-sight stability and are specifically engineered for precision optical tracking. In contrast, RSONAR was flown on a CNES/CSA STRATO Science gondola, whose attitude control system is de- signed for coarse pointing at the sub-degree level rather than fine stabilization. The attitude estimates used in this work were derived from 27-frame stacked images, which provide robust star solutions but inherently smooth short-timescale motion. Using single-frame attitude solutions would yield more precise instantaneous pointing estimates, but doing so would increase the computational load by roughly a factor of 26 and was therefore not practical for the present analysis.”

Reviewer 2 Report

Comments and Suggestions for Authors

An interesting manuscript with a detailed description of the capabilities of a start tracker in space debris observations. After some modifications I can recommend to publish your manuscript in this Journa.

Major comment:

all references appear with question marks. Please, fix them.

Minor comments:

Although it is clear that LEO is low-earth orbit here, but please, resolve all abbreviation at the first use. Or, please put the list of the abbreviations from the end of the paper to the beginning of it.

Line 34: what is hypervelocity? Please define it.

Line 64: I think the capabilities of optical telescopes are not limited only by clouds, but also by moonlight and/or light pollution. 

Line 87-92: since the references are not given (because all references are denoted by a question mark), I cannot check the calculation of the prices given in the manuscript. How did they were calculated? What lifetime of a sensor was assumed?

line 98: define ST here (I guess it s Star Tracker)

line 114: explain Fried parameter in a footnote or give a reference to it and for the statement, please

Table 1.: The regular name of the !F-number! is focal ratio, please write this

Table 1: if 'chromaticity' is monochrome, then what wavelengths were used? Or do the authors mean white light observations, no filter?

Sections 2 and 3.: how precisely was the position of the gondola known? How did the gondola determine its position (geographical latitude, longitude and height over the sea level)?

Section 3.4: how did you subtract the sky-background from the fluxes measured by aperture photometry; Or, was the sky background negligible at this short exposures?

Figure 6: what causes the cloud of points roght and below from the main trend and the fit? Are they unresolved binary stars?

Figure 8: can you give, please, the RA, DEC coordinates of the image center and the observation date and time in the figure caption?

Author Response

We would like to thank the reviewers for their careful reading and constructive comments. All suggested changes have been implemented in the revised manuscript, and highlighted text indicates the edits made in response to the reviews.

Below we address each comment point-by-point.

Major comment:

all references appear with question marks. Please, fix them.

All broken citations have been corrected and now render properly throughout the manuscript.

Minor Comments

Comment 1: Although it is clear that LEO is low-earth orbit here, but please, resolve all abbreviation at the first use. Or, please put the list of the abbreviations from the end of the paper to the beginning of it.

Response: We have ensured that abbreviations are resolved at first use. For example, LEO is now defined in the Introduction (line 30):

“The Low Earth Orbit (LEO) region remains the most congested, containing 57% of monitored RSOs, with 9,016 space debris originating from rockets and payloads [3].”

We have also updated and expanded the Abbreviations section to include all acronyms used in the manuscript, such as LEO, MEO, GEO, ASAT, sCMOS, FPGA, WCS, ICRS, TRL, NMEA, MAD, LoA, and CNES.

Comment 2: “Line 34: what is hypervelocity? Please define it.”

Response: We agree and have clarified the meaning of “hypervelocity” in the context of LEO by adding a definition and citation. Line 34 now reads:

“These objects travel at hypervelocities, typically 7–8 km/s in LEO [4], posing a significant threat to operational satellites [5].”

Comment 3: “Line 64: I think the capabilities of optical telescopes are not limited only by clouds, but also by moonlight and/or light pollution.”

Response: We thank the reviewer for this important point. In Section 1.1, we expanded the discussion of environmental limitations on optical telescopes. Lines 65–67 now state:

“However, the operational period of these telescopes is limited by cloud coverage, and they are only functional during the night. These systems are also susceptible to light pollution, sky brightness and exhibit reduced performance when imaging RSOs located within approximately 15° of the Moon’s center [19].”

Comment 4: “Line 87–92: since the references are not given (because all references are denoted by a question mark), I cannot check the calculation of the prices given in the manuscript. How did they were calculated? What lifetime of a sensor was assumed?”

Response: We appreciate this comment and have clarified the cost model and sensor lifetimes used. As described in Ackermann et al. (2015) [20], cost-per-observation is computed by annualizing the acquisition cost of each sensor over its assumed operational lifetime and dividing by the total number of observations acquired per year. Their model assumes 40-year lifetimes for ground-based optical telescopes, 7 years for the GeOST/ORS-5 system, and 10 years for Sapphire-class SSA spacecraft, with annual operations and maintenance costs equal to 5–10% of the capital cost.

We have added the following clarifying sentence to Section 1.1 (lines 92–93):

“These values follow the cost model in [20], which annualizes system acquisition cost over the assumed lifetimes (40 years for ground-based, 7–10 years for space-based) and divides by the yearly number of observations.”

Comment 5: “Line 98: define ST here (I guess it’s Star Tracker).”

Response: We have defined the ST abbreviation at first use. Line 101 now reads:

“Additionally, Star Trackers (ST) are traditionally employed for attitude determination and have been investigated for RSO imaging [22,23].”

Comment 6: “Line 114: explain Fried parameter in a footnote or give a reference to it and for the statement, please.”

Response: We have added a footnote and reference defining the Fried parameter on page 3 (line 117):

“The Fried parameter is a measure of the atmospheric coherence length and quantifies the strength of optical turbulence [26].”

Comment 7: “Table 1.: The regular name of the F-number is focal ratio, please write this.”

Response: We have replaced “F-number” with “Focal Ratio” in Table 1 and updated the abbreviation definition accordingly in the Abbreviations section.

Comment 8: “Table 1: if 'chromaticity' is monochrome, then what wavelengths were used? Or do the authors mean white light observations, no filter?”

Response: Thank you for pointing out this ambiguity. In Table 1, we now explicitly state:

“Chromaticity: Monochrome (unfiltered)”

to clarify that the observations were conducted without an optical filter over the native spectral response of the sensor.

Comment 9: “Sections 2 and 3.: how precisely was the position of the gondola known? How did the gondola determine its position (geographical latitude, longitude and height over the sea level)?”

Response: We have added a detailed explanation in Section 3.1 (Attitude Determination) (lines 221–227):

“Gondola position and state information were provided by the CSA μPRISM avionics module. μPRISM supplied GPS latitude, longitude, and altitude at a 1 Hz rate through standard NMEA sentences, with differential-GPS refinement used for attitude and heading. The geodetic coordinates were reported to four decimal places in latitude and longitude (corresponding to a numerical resolution of ≈10–12 m at mid-latitudes) and 1 m in altitude. These GPS measurements were used directly as the observer position for all subsequent astrometric and photometric transformations.”

Comment 10: “Section 3.4: how did you subtract the sky-background from the fluxes measured by aperture photometry; Or, was the sky background negligible at this short exposures?”

Response: We thank the reviewer for this question. We clarify that each flux value used in the photometric analysis is background-subtracted exactly once.

For stellar fluxes, the values returned by Astrometry.net are already background-subtracted by its internal source-extraction routine and are used directly for stellar calibration.

For RSO fluxes, measurements are performed on raw single-frame images using aperture–annulus photometry, with a concentric background annulus to estimate and subtract the local sky background. This procedure is described in Section 3.4.1.

To remove any ambiguity, we added the following clarification in Section 3.4 (Photometric Residuals) (lines 257–260):

“The flux values returned in the Astrometry.net output tables are computed by its internal source-extraction algorithm, which estimates and subtracts the local sky background before integrating the object signal. These background-subtracted stellar fluxes were therefore used directly in the photometric calibration.”

Comment 11: “Figure 6: what causes the cloud of points right and below from the main trend and the fit? Are they unresolved binary stars?”

Response: We have added an expanded discussion in Section 5.4 (Photometric Residuals) (lines 569–580):

“A small secondary cloud of points below the main flux–magnitude relation (Fig. 6) appears in a limited number of stacked sequences. These points are associated with brief intervals of increased gondola motion, which reduce the recovered stellar flux in the stacks and cause some faint stars to fall below the primary calibration trend. The same mechanism explains the ‘wings’ observed in the Bland–Altman comparison. During intervals of elevated gondola motion, PSF broadening and slight frame-to-frame misregistration decrease the stellar flux used in the calibration transfer. Because the stellar calibration is applied directly to single-frame RSO photometry, these flux underestimates manifest as larger physics–empirical magnitude differences. Such discrepancies are therefore observational and processing-related rather than intrinsic limitations of the reflectance model. Future analyses could mitigate these effects through motion-aware stacking, PSF-shape diagnostics, and per-image weighting of calibration frames.”

Comment 12: “Figure 8: can you give, please, the RA, DEC coordinates of the image center and the observation date and time in the figure caption?”

Response: We have revised the Figure 8 caption to include the image-center RA and Dec, field of view, and observation times for both the single-frame and stacked images. For example, the caption now specifies:

“A single 0.1 s exposure acquired at 05:04:31.758 UTC, centered at (RA, Dec) = (344.365°, 16.051°) with a 29.7° × 29.7° field of view…” and similarly for the stacked composite.

Reviewer 3 Report

Comments and Suggestions for Authors

This manuscript presents observations of SSA targets using a dual-purpose star tracker installed on the RSONAR stratospheric balloon. The study details the data processing methods, results, and discussions related to instrument parameters, attitude determination, full width at half maximum, astrometric, and photometric performance. It demonstrates the feasibility of employing a dual-purpose star tracker for space situational awareness and environmental monitoring.

Overall, the manuscript provides a thorough background introduction, detailed data processing and analysis, and an in-depth discussion of the results, which strongly support the conclusions. With the following revisions, it will be suitable for publication.

  1. The statistically negligible correlation in Figure 5 could be strengthened by calculating the Pearson correlation coefficient. I recommend including this value for all subplots in Figure 5 and for the upper-right panel in Figure 6.
  2. A minor typo ("studdy" for "study") was found on line 402 and should be amended.

Author Response

We would like to thank the reviewers for their careful reading and constructive comments. All suggested changes have been implemented in the revised manuscript, and highlighted text indicates the edits made in response to the reviews.

Below we address each comment point-by-point.

Comment 1: The statistically negligible correlation in Figure 5 could be strengthened by calculating the Pearson correlation coefficient. I recommend including this value for all subplots in Figure 5 and for the upper-right panel in Figure 6.

Response: We thank the reviewer for this suggestion. The Pearson correlation coefficients have now been added to each panel of Figure 5, and to the upper-right panel of Figure 6.

In Section 4.3 (Astrometric Residuals), we added the following explanation (lines 396–400):

“Pearson correlation coefficients are reported in each panel of Figure 5. For the first five intervals, the coefficients lie between −0.12 and +0.07, confirming that the residuals are statistically uncorrelated, while the larger coefficient in the final interval (r = −0.634) reflects the much smaller number of available residuals (n = 1815) in that six-minute segment. A slight broadening appears in the final interval (05:26–05:33 am).”

In Section 5.3 (Astrometric Residuals), we further clarified the interpretation (lines 545–551):

“The slight elongation of the residual cloud is consistent with the increased gondola motion near sunrise and does not indicate a change in the underlying astrometric solution. This interpretation is supported by the negligible Pearson correlations in the first five intervals, which demonstrate that RA and Dec residuals are not systematically coupled and that the astrometric calibration remained unbiased.”

For Figure 6, the Pearson correlation coefficient between field radius and photometric residuals has been added (upper-right panel and caption), and the following sentence has been included in Section 4.4 (Photometric Residuals) (lines 407–409):

“The Pearson correlation coefficient between field radius and photometric residuals was r = +0.018, indicating that the residuals are statistically uncorrelated with detector position.”

Comment 2: A minor typo ("studdy" for "study") was found on line 402 and should be amended.

Response: Thank you for spotting this. The typo has been corrected in the revised manuscript.